# PROTOTYPE DECODING AND RE-ENCODING FOR CLASS-INCREMENTAL LEARNING

## ABSTRACT

Prototype-based Class-Incremental Learning (CIL) methods achieve competitive performance to mitigate catastrophic forgetting, requiring only the storage of prototypes in memory instead of retaining past task exemplars. However, as the encoder updates during CIL sessions, prototypes computed in earlier sessions by the old encoder can become outdated, i.e., mismatching the accordingly drifted ground-truth prototypes of past task data. In this paper, we propose a prototype update mechanism, termed Prototype Decoding and Re-encoding (PDR). We found that by combining the knowledge of stored prototypes and the latest frozen encoder, we can obtain a high-quality approximation of the sample distribution of past data, and use this extra information to guide the prototype update. Our prototype update mechanism is plug-and-play and can be seamlessly integrated with most prototype-based CIL methods. A computationally efficient three-step PDR stage is added after the encoder is trained by a prototype-based CIL method. First, we use the old encoder and stored prototypes to guide the decoder training. Then, an adapter is trained by the pseudo exemplars generated by the decoder. Finally, we use the adapter to re-encode the stored prototypes. Extensive experiments show that a simple CIL method LDC combined with PDR outperforms current exemplar-free baselines by up to 3.86%. Moreover, the inclusion of the PDR mechanism results in no additional time overhead per session, relative to the already time-efficient LDC baseline.

## 1 INTRODUCTION

Deep Neural Networks (DNNs) (Larochelle et al., 2009) have achieved remarkable success across a wide spectrum of visual recognition tasks. Nevertheless, the conventional supervised learning paradigm assumes that all training data for every task is accessible in one session, a scenario that rarely aligns with real-world applications. Class-Incremental Learning (CIL) addresses this gap by enabling models to progressively integrate new classification tasks as they arrive (Kirkpatrick et al., 2017). In CIL, the network is exposed to a sequence of tasks, each introducing new classes, while it must continue to accurately classify exemplars from all previously encountered classes. Although the human brain seamlessly integrates new information without discarding old knowledge, DNNs often struggle with catastrophic forgetting (Li & Hoiem, 2017)–an abrupt loss of previously acquired skills when learning new tasks. Thus, effective CIL methods must balance model plasticity, to assimilate information of new tasks, with stability, to avoid catastrophic forgetting of past tasks.

A widely adopted solution to mitigate forgetting involves exemplar-based methods (Aljundi et al., 2019; Bang et al., 2021; Zhuo et al., 2023; Boschini et al., 2022; Buzzega et al., 2021), which retain a limited set of representative exemplars from each class in a memory buffer. Despite their effectiveness, these methods are highly contingent on the quality of stored exemplars and face challenges related to storage overhead and privacy problems that are especially acute in sensitive applications like medical imaging (Shi & Ye, 2024). This has spurred interest in exemplar-free methods (Kirkpatrick et al., 2017; Zenke et al., 2017; Li & Hoiem, 2017; Zhu et al., 2021; Magistri et al., 2024), which avoid retaining raw past data. Prototype-based methods (Zhu et al., 2022; 2021; Petit et al., 2023; Goswami et al., 2024; Magistri et al., 2024), as leading exemplar-free methods, store class prototypes–often computed as the class means in the deep feature space–to model the feature distribution of training data and inject information from the past classes in subsequent CIL sessions. However, as the encoder updates during CIL sessions, prototypes computed in earlier sessions by the

old encoder can become outdated, i.e., mismatching the accordingly drifted ground-truth prototypes of past task data. Without access to past task data, as is the case in exemplar-free scenarios, this mismatch cannot be properly corrected, leading to failure, especially in long task sequence conditions and when the amount of early task data is low. Few works attempt to mitigate this problem. Gomez-Villa et al. (2024) train an adapter to learn the mapping of the representation changes of the current task data from the old encoder to the new encoder, and then uses this adapter to map the stored prototypes to the new space to make it more consistent with the ground-truth prototypes. However, the representation mapping of the current task data cannot fully represent the representation mapping of the past task data, and the stored prototypes correspond to the past task data. Therefore, the stored prototypes updated using this method will have an increasing mismatch with the ground-truth prototypes.

To solve this problem, we consider decoding prototypes to obtain pseudo exemplars of past tasks, and then learning an adapter based on the representation mapping of these pseudo exemplars to update prototypes. In this paper, we propose a prototype update mechanism, termed Prototype Decoding and Re-encoding (PDR). We found that by combining the knowledge of stored prototypes and the latest frozen encoder, we can obtain a high-quality approximation of the sample distribution of past data, and use this extra information to guide the prototype update. Our prototype update mechanism is plug-and-play and can be seamlessly integrated with most prototype-based CIL methods. Specifically, our proposed PDR mechanism adds a time-efficient stage to update the prototypes after the encoder is trained by a prototype-based CIL method. This stage includes three steps: training decoder, training adapter, and re-encoding prototypes. First, a decoder is trained to synthesize pseudo exemplars, which are later used for adapter training. The decoder takes as input noise vectors sampled from a Gaussian distribution and outputs pseudo exemplars that serve as proxies for data from previous tasks. During decoder training, the generated pseudo exemplars are passed through the old encoder to obtain pseudo representations. The decoder is optimized to ensure that each pseudo representation is close to the stored prototype of its category. Then, the adapter is trained by aligning the representations produced by the old and new encoders across all inputs, including both generated pseudo exemplars and current task exemplars. Finally, the stored prototypes are fed into the adapter to be re-encoded into updated prototypes, which are then stored in memory.

We conduct extensive experiments on multiple benchmarks, and the results show that integrating our proposed PDR mechanism with existing CIL methods consistently improves performance across both small- and large-scale datasets. Specifically, when combined with PDR, the LDC method outperforms its variant without PDR and surpasses state-of-the-art exemplar-free approaches, achieving up to a 3.86% improvement on CIFAR-100, 3.35% on TinyImageNet, and 3.65% on ImageNet-100 compared to the best baseline. Moreover, PDR demonstrates strong adaptability across different scenarios, delivering significant gains in both short sequences and long sequences, whether the early tasks contain many classes (Warm Start) or only a few classes (Cold Start). Notably, despite the performance gains, the additional PDR stages result in no additional time overhead per session when compared to the already highly efficient LDC baseline. This is because the decoder training is independent of the encoder training, allowing both processes to be executed in parallel, and the training adapter and re-encoding prototypes steps incur only negligible overhead, as these steps are also present in LDC. In addition, unlike deep generative replay (Shin et al., 2017; Gao & Liu, 2023), PDR does not introduce any extra memory cost for prototype-based methods, as the decoder and adapter are discarded at the end of each session, leaving only the class prototypes to be stored.

In summary, our contributions are as follows. 1) We propose to decode the prototypes to obtain pseudo exemplars that better match the distribution of past task data by mining the information at the sample space level in the prototypes. 2) We propose re-encoding the stored prototypes by an adapter trained on generated pseudo exemplars to mitigate the mismatch between stored prototypes and ground-truth prototypes. 3) By integrating the above two components, we propose a novel prototype update mechanism, which consistently improves the performance of the existing prototype-based continual learning method across a wide range of benchmarks.

## 2    RELATED WORKS

The methods of Class-Incremental Learning (CIL) can be divided into two categories according to whether they need to store exemplars. Exemplar-based methods mitigate catastrophic forgetting by

replaying a limited set of representative samples of old classes from a memory buffer (Aljundi et al., 2019; Bang et al., 2021; Buzzega et al., 2020; Boschini et al., 2022). For instance, Experience Replay (ER) (Rebuffi et al., 2017) involves interleaving old samples with current data in training batches. Dark Experience Replay (DER) (Buzzega et al., 2020) and eXtended-DER (X-DER) (Boschini et al., 2022) match the network's logits sampled throughout the optimization trajectory, thus promoting consistency with its past.

The privacy leakage and storage cost issues of exemplar-based methods have spurred interest in exemplar-free methods, including architecture-based methods (Rusu et al., 2016; Mallya & Lazebnik, 2018; Serra et al., 2018), regularization-based (Kirkpatrick et al., 2017; Zenke et al., 2017), pseudo replay-based methods (Li & Hoiem, 2017; Gao et al., 2022; Smith et al., 2021), and prototype-based methods (Zhu et al., 2022; 2021; Petit et al., 2023; Goswami et al., 2024; Magistri et al., 2024). Regularization-based methods gather gradient information to estimate the importance of weights, followed by devising a loss term to prevent significant deviations from the previous configuration. Architecture-based Methods allocate specific sets of model parameters to different tasks. Pseudo-replay-based methods use the current data samples (Li & Hoiem, 2017) or samples generated by model inversion (Gao et al., 2022) as the proxy of replay exemplars.

Prototype-based methods, which have received increasing attention in recent years, showed the strongest performance in exemplar-free methods. Prototype-based methods store class-representative prototypes in memory and utilize them to enhance the classifier performance and mitigate catastrophic forgetting. PASS (Zhu et al., 2021) aligns prototypes of newly introduced classes with those from prior tasks by applying noise-based augmentation. SSRE (Zhu et al., 2022) emphasizes progressively enriching the model's representation space to integrate new classes effectively. FeTrIL (Petit et al., 2023) generates pseudo-features for previously learned classes by transforming old prototype features based on the shifts between old and new prototypes in the feature space. FeCAM (Goswami et al., 2024) leverages the heterogeneous feature distribution inherent to class-incremental learning by employing an anisotropic Mahalanobis distance metric, which offers a more nuanced alternative to the conventional Euclidean distance. However, none of the above methods address the mismatch between stored prototypes and ground-truth prototypes. LDC (Gomez-Villa et al., 2024) proposes to train an adapter to learn the feature mapping of current task data, and uses this adapter to update the stored prototypes. However, the feature mapping of current task data cannot fully represent the feature mapping of past task data.

## 3 PRELIMINARIES

### 3.1 PROBLEM FORMULATION AND NOTATIONS

In CIL, we are given a series of classification tasks, with each task presented sequentially. Each task $t$ (also be denoted as session $t$ in the following paper) is associated with a training dataset $\mathcal{D}_t = \{\mathcal{X}_t, \mathcal{Y}_t\}$ and a test dataset, and no two tasks share any categories in common. The set of categories in task $t$ is denoted as $\mathcal{C}_t$, and the set of categories in all task $1$ to $t$ is denoted as $\mathcal{C}_{1:t}$. At session $t$, the classification model is composed of the encoder $f(\cdot; \theta_t)$ shared across all tasks and whose parameters $\theta_t$ are updated during training, and a linear classifier $W_t \in \mathbb{R}^{d \times |\mathcal{C}_{1:t}|}$, which grows with each new task. $d$ denotes the dimension of the features output by the encoder $f(\cdot; \theta_t)$. The model output at task $t$ is the composition of encoder and classifier, i.e., $\mathrm{softmax}(W_t^\top f(x; \theta_t))$. The goal of CIL is to design a strategy for updating the model such that the final model at time $t$ performs well on all test data from all seen tasks $1$ to $t$. Exemplar-free CIL methods will not store any samples of past data in memory for more memory efficiency and privacy protection, so some of them, prototype-based methods, store prototypes in memory instead.

### 3.2 PROTOTYPE-BASED CIL METHODS

In prototype-based CIL methods, at the end of session $t$, for each category $c$ in current task, the mean of the features of all samples extracted by the newest encoder is stored as the prototype $p^c$, that is $p^c = \frac{1}{|\mathcal{X}^c|} \sum_{x \in \mathcal{X}^c} f(x; \theta_t)$, where $\mathcal{X}^c$ is the images of category $c$. At the beginning of session $t$, all prototypes of learned categories $\mathcal{P}_{1:t-1} = \{p^c | c \in \mathcal{C}_{1:t-1}\}$ are stored in memory. Then prototype-based methods use these stored prototypes to guide the training at session $t$.

## 4 METHODOLOGY

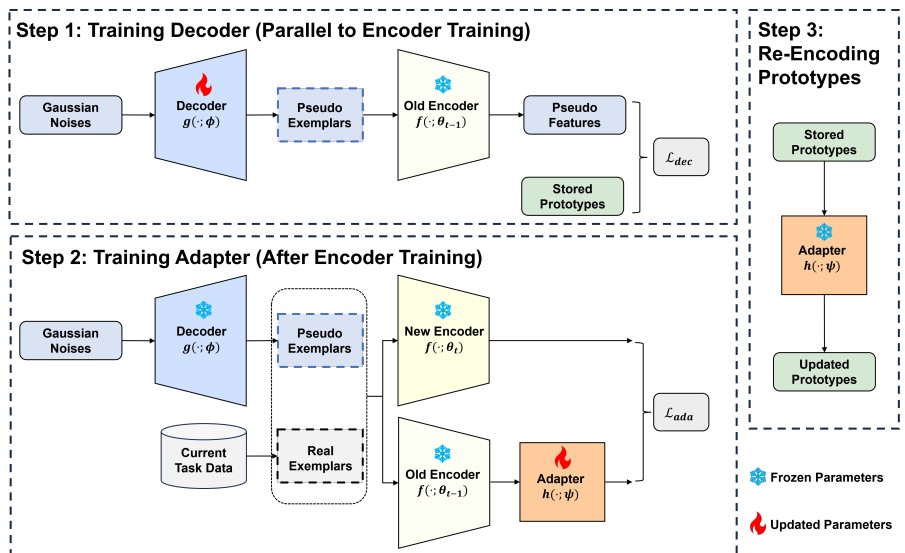

Figure 1: Overall framework of Prototype Decoding and Re-encoding.

An illustration of our framework is shown in Figure 1. At the beginning of session $t$ ($t > 1$), the contents in the storage include the latest encoder $f(\cdot; \theta_{t-1})$ and classifier $W_{t-1}$ trained in previous sessions and the stored prototypes $\mathcal{P}_{1:t-1}$ updated in the last session. In each session, first, the encoder is trained by a prototype-based CIL method. Then our proposed PDR mechanism adds three time-efficient steps to update the prototypes after the encoder training, including training decoder, training adapter, and re-encoding prototypes.

**Training Decoder.** Decoder training is independent of the encoder training, allowing both processes to be executed in parallel. We consider using prototypes to reconstruct the sample distribution of old classes and generate pseudo exemplars that serve as proxies for past task exemplars.

Only using stored prototypes makes it hard to reconstruct the past sample distribution. The model inversion technique (Yin et al., 2020) demonstrates that using the model's encoder and classifier, a generator can be trained to sample pseudo exemplars that approximate the model's training data. However, the pseudo exemplars trained by model inversion have a gap with the real exemplars. Thus, using these pseudo exemplars in CIL leads to poor performance (Smith et al., 2021). In this paper, we combine the information of stored prototypes with the model inversion technique.

The decoder $g(\cdot; \phi)$ is a model with parameters $\phi$. The decoder's inputs are Gaussian noises, and its outputs are pseudo exemplars $\tilde{\mathcal{X}}_{1:t-1}$ that can serve as proxies for real exemplars in all past tasks.

During decoder training, the number of sampled noises from a Gaussian distribution is $N = \sum_{c \in \mathcal{C}_{1:t-1}} n^c$, where $n^c$ is the number of samples of category $c$. A category is uniformly sampled from $\mathcal{C}_{1:t-1}$ for each noise as the label of this noise. Then these noises will be input to the decoder $g(\cdot; \phi)$ to generate $N$ pseudo exemplars. We use the frozen encoder $f(\cdot; \theta_{t-1})$ to extract $N$ pseudo features from these pseudo exemplars. We will calculate two learning objectives and optimize only $g(\cdot; \phi)$ based on these objectives.

The first objective is batch normalization regularization, a commonly used objective in model inversion (Yin et al., 2020), to constrain the mean and variance of the feature at the BatchNorm layer to be consistent with the running-mean and running-variance of the same layer, that is

$$\mathcal{L}_{bn} = \sum_{l} \mathcal{L}_{kl}(\mathcal{N}(\sigma_l, \mu_l), \mathcal{N}(\hat{\sigma}_l, \hat{\mu}_l)),$$

where $l$ indicates the $l$-th BatchNorm layer in the model, $\sigma_l$ and $\mu_l$ denote the mean and variance of the $l$-th BatchNorm layer stored in the frozen encoder, $\hat{\sigma}_l$ and $\hat{\mu}_l$ denote the mean and variance estimated on the pseudo exemplars, and $\mathcal{L}_{kl}$ denotes the KL divergence.

Most of the model inversion methods construct the second objective based on the assumption that each pseudo exemplar can be easily classified by the encoder and classifier into its true category. Thus, these methods introduce a cross-entropy loss as the second objective. However, we found that a prototype-based loss can better guide the pseudo exemplars to match the distribution of the real exemplars. The prototype-based loss is

$$\mathcal{L}_{pro} = \frac{1}{|\tilde{\mathcal{D}}_{1:t-1}|} \sum_{(\tilde{x},\tilde{y}) \in \tilde{\mathcal{D}}_{1:t-1}} \mathcal{L}_{ce}(\text{sim}(\mathcal{P}_{1:t-1}, f(\tilde{x}; \theta_{t-1})), \tilde{y}),$$

where $\text{sim}(\mathcal{P}_{1:t-1}, f(\tilde{x}; \theta_{t-1}))$ denotes a vector with dimension $|\mathcal{C}_{1:t-1}|$ that the $i$-th element is the cosine similarity between the $i$-th prototype in $\mathcal{P}_{1:t-1}$ and the pseudo feature $f(\tilde{x}; \theta_{t-1})$. This loss creates attractive forces between $f(\tilde{x}; \theta_{t-1})$ and the prototype of its category and repulsive forces between $f(\tilde{x}; \theta_{t-1})$ and all other prototypes.

We use $t$-SNE visualization results to show that pseudo exemplars trained using prototype-based loss match real samples better than classifier-based loss at the representation level. In Figure 2, we compare the pseudo features generated by PDR to the pseudo features generated by two other methods. The circles, squares, and pentagons are the generated pseudo representations of the three methods. The forks in the three figures all refer to the ground-truth distribution of the real data in the current feature space. The figures of the same color represent the same category. The numbers represent the positions of the prototypes of the corresponding labeled categories. The $t$-SNE visualization results show that our generated pseudo features (pentagons in Figure 2c) best match the ground-truth features. In contrast, the pseudo features generated by traditional model inversion with classifier-based loss (squares in Figure 2b) are not well-split between categories. The pseudo features generated by adding Gaussian noises on prototypes (circles in Figure 2a), utilized in many SOTA prototype-based methods such as PASS (Zhu et al., 2021), have too little diversity compared to the ground-truth features.

The total training objective is $\mathcal{L}_{dec} = \mathcal{L}_{bn} + \alpha \mathcal{L}_{pro}$, where $\alpha$ is the weight to balance the two objectives of decoder training.

**Training Adapter**  When the prototypes are first calculated, they can represent the distribution of each class in the feature space. However, as the number of training sessions increases, the encoder is constantly updated, causing the prototypes calculated in the past to increasingly deviate from the distribution of the real data of each category in the current feature space.

Therefore, at the end of each session, for classes of past tasks, we propose to use the synthetic pseudo exemplars in the current session and the current task data to train an adapter network $h(\cdot; \psi)$ with parameters $\psi$ to re-encode the prototypes. We minimize the mean squared error (MSE) between the extracted features by old encoder and new encoder. Hence, the objective to train $h(\cdot; \psi)$ is

$$\mathcal{L}_{ada} = \frac{1}{|\tilde{\mathcal{X}}_{1:t-1} \cup \mathcal{X}_t|} \sum_{x \in (\tilde{\mathcal{X}}_{1:t-1} \cup \mathcal{X}_t)} ||h(f(x; \theta_{t-1}) - f(x; \theta_t); \psi)||^2,$$

where $f(\cdot; \theta_{t-1})$ and $f(\cdot; \theta_t)$ are frozen.

**Re-Encoding Prototypes.**  Finally, the prototypes of old tasks are updated by

$$p^c \leftarrow h(p^c; \psi), \quad \forall \, p^c \in \mathcal{P}_{1:t-1}.$$

For classes of the current task, we use the real exemplars to calculate the prototypes and append them to the stored prototypes.

## 5  EXPERIMENTS

### 5.1  EVALUATION PROTOCOL

**Datasets.**  Same as previous works (Zhu et al., 2021; 2022; Magistri et al., 2024; Gomez-Villa et al., 2024), we evaluated our approach on CIFAR100 (Krizhevsky et al., 2009), TinyImageNet (Le & Yang, 2015) and ImageNet100 (Deng et al., 2009). We split CIFAR100 into 5 tasks (20

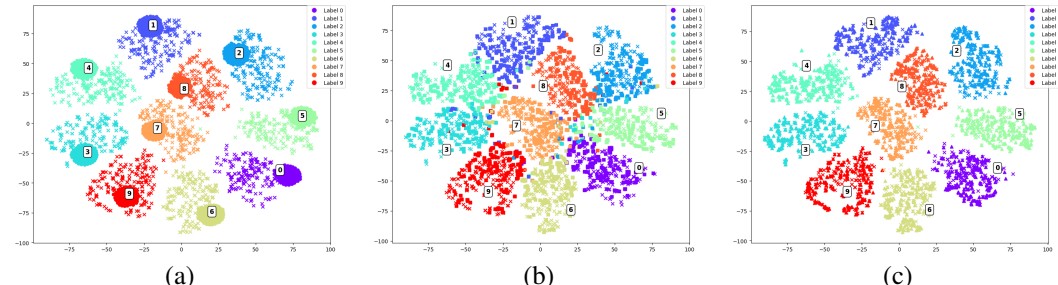

(a)                 (b)                 (c)

Figure 2: $t$-SNE Visualization of Feature Space. We train a classification model in the first session of CIFAR100 Cold Start $T = 10$ and generate pseudo features of the categories of the first task by three different methods at the beginning of the second session. The circles in (a) are pseudo features generated by adding Gaussian noises on prototypes. The squares in (b) are pseudo features generated by traditional model inversion. The pentagons in (c) are pseudo features generated by prototype decoding (our method). The forks in the three figures all refer to the ground-truth distribution of the real data in the current feature space. The figures of the same color represent the same category. The numbers represent the positions of the prototypes of the corresponding labeled categories.

categories per task), 10 tasks (10 categories per task), and 20 tasks (5 categories per task). We split TinyImageNet into 5 tasks (40 categories per task), 10 tasks (20 categories per task), and 20 tasks (10 categories per task). We split ImageNet100 into 5tasks (20 categories per task), 10 tasks (10 categories per task), and 20 tasks (5 categories per task). ImageNet100 is a subset of ImageNet dataset, which is widely used as the benchmark in most of the previous CIL methods (Zhu et al., 2021; 2022; Magistri et al., 2024; Gomez-Villa et al., 2024).

Our method is evaluated in both more challenging *Cold Start* scenarios and *Warm Start* scenarios. We primarily focus on Cold Start, where the encoder undergoes more drastic updates, making the issue of prototype obsolescence more pronounced. Most previous prototype-based methods have been evaluated in Warm Start scenarios (Zhu et al., 2021; 2022)–where the first task includes a disproportionately large number of classes (typically 40–50% of the total dataset). This is because they need more classes in the first session to learn a robust encoder, and the prototypes computed by this encoder will not be drastically outdated in the future sessions (Gomez-Villa et al., 2024). In contrast, in the more challenging Cold Start scenarios (Buzzega et al., 2020; Gomez-Villa et al., 2024), the initial task is limited, necessitating a reduction in the discrepancy between stored and true prototypes.

**Metrics.** In line with previous works (Boschini et al., 2022; Zhu et al., 2021; Gao et al., 2022; Wang et al., 2025; Magistri et al., 2024), we report two metrics of each setting, including $A_{last}$ and $A_{inc}$. We denote the test accuracy of task $i$ after training task $j$ as $A_i^j$, and the total number of tasks is $T$. $A_{last}$ is the average accuracy on all seen tasks by the model after all sessions, and $A_{inc}$ is computed by calculating each average accuracy of the model trained after each session on all seen tasks, and averaging all the accuracy, that is

$$A_{last} \triangleq \frac{1}{T} \sum_{t=1}^{T} A_t^T, \quad A_{inc} \triangleq \frac{1}{T} \sum_{t=1}^{T} \frac{1}{t} \sum_{s=1}^{t} A_s^t.$$

Higher $A_{last}$ and $A_{inc}$ indicate better performance.

**Implementation details.** Following prior works (Boschini et al., 2022; Zhu et al., 2021; Gao et al., 2022; Wang et al., 2025; Magistri et al., 2024), we employ a ResNet18 architecture (He et al., 2016), initialized from scratch (i.e., not pre-trained), for the encoder $f(\cdot; \theta)$. The class-conditional decoder $g(\cdot; \phi)$ takes as input a noise vector $z \in \mathbb{R}^{512}$ combined with a class embedding $e_y \in \mathbb{R}^{512}$, projects it to a $4 \times 4$ spatial seed with 512 channels, and then applies four upsampling residual blocks with channel widths $512 \rightarrow 256 \rightarrow 128 \rightarrow 64$, followed by a final $3 \times 3$ convolution to generate RGB images. For training the decoder $g(\cdot; \phi)$, we use the ADAM optimizer (Kingma & Ba, 2014) with a learning rate of $0.002$. The decoder is trained for 20 epochs using a batch size of 400. The hyperparameter $\alpha$ is set to $0.05$. For training the feature encoder (the ResNet18 backbone within

Table 1: Evaluation on CIFAR100 with protocol that equally splits classes into $T$ tasks.

| Method | $T=5$ | | $T=10$ | | $T=20$ | |
|---|---|---|---|---|---|---|
| | $A_{last}$ (%) | $A_{inc}$ (%) | $A_{last}$ (%) | $A_{inc}$ (%) | $A_{last}$ (%) | $A_{inc}$ (%) |
| EWC (Kirkpatrick et al., 2017) | - | - | 31.17 ± 2.94 | 49.14 ± 1.28 | 17.37 ± 2.43 | 31.02 ± 1.15 |
| LwF (Li & Hoiem, 2017) | - | - | 32.80 ± 3.08 | 53.91 ± 1.67 | 17.44 ± 0.73 | 38.39 ± 1.05 |
| ABD (Smith et al., 2021) | 46.79 ± 0.48 | 63.16 ± 1.49 | 37.01 ± 0.93 | 57.32 ± 1.93 | 22.14 ± 0.65 | 44.53 ± 2.01 |
| R-DFCIL (Gao et al., 2022) | 49.90 ± 0.43 | 64.78 ± 1.78 | 42.57 ± 0.72 | 59.13 ± 1.76 | 30.35 ± 0.12 | 47.80 ± 1.90 |
| CCIL (Wang et al., 2025) | 52.46 ± 0.35 | 66.31 ± 1.46 | 43.69 ± 0.57 | 60.14 ± 1.87 | 31.72 ± 0.22 | 49.01 ± 1.81 |
| SSRE (Zhu et al., 2022) | - | - | 30.40 ± 0.74 | 47.26 ± 1.91 | 17.52 ± 0.80 | 32.45 ± 1.07 |
| PASS (Zhu et al., 2021) | - | - | 30.45 ± 1.01 | 47.86 ± 1.93 | 17.44 ± 0.73 | 32.86 ± 1.03 |
| FeTrIL (Petit et al., 2023) | - | - | 34.94 ± 0.46 | 51.20 ± 1.13 | 23.28 ± 1.24 | 38.48 ± 1.07 |
| LDC (Gomez-Villa et al., 2024) | 59.19 ± 0.02* | 71.23 ± 1.61* | 45.40 ± 2.80 | 59.50 ± 3.90 | 36.38 ± 0.76* | 54.30 ± 0.44* |
| **LDC+PDR** | **60.53±0.21** | **71.80±0.49** | **48.14±0.95** | **64.00±2.07** | **39.80±0.17** | **56.70±2.43** |

Table 2: Evaluation on TinyImageNet with the protocol that equally splits classes into $T$ tasks.

| Method | $T=5$ | | $T=10$ | | $T=20$ | |
|---|---|---|---|---|---|---|
| | $A_{last}$ (%) | $A_{inc}$ (%) | $A_{last}$ (%) | $A_{inc}$ (%) | $A_{last}$ (%) | $A_{inc}$ (%) |
| EWC (Kirkpatrick et al., 2017) | - | - | 8.00 ± 0.27 | 24.01 ± 0.51 | 5.16 ± 0.54 | 15.70 ± 0.35 |
| LwF (Li & Hoiem, 2017) | - | - | 26.09 ± 1.29 | 45.14 ± 0.88 | 15.02 ± 0.67 | 32.94 ± 0.54 |
| ABD (Smith et al., 2021) | 30.40 ± 0.78 | 45.07 ± 0.78 | 22.50 ± 0.62 | 40.52 ± 0.71 | 15.65 ± 0.95 | 35.00 ± 0.53 |
| R-DFCIL (Gao et al., 2022) | 35.25 ± 0.57 | 48.90 ± 1.03 | 29.96 ± 0.36 | 44.58 ± 0.68 | 24.07 ± 0.28 | 39.06 ± 0.62 |
| CCIL (Wang et al., 2025) | 36.89 ± 0.73 | 49.69 ± 0.76 | 30.90 ± 0.52 | 45.29 ± 0.73 | 24.54 ± 0.27 | 39.65 ± 0.19 |
| SSRE (Zhu et al., 2022) | - | - | 22.93 ± 0.95 | 38.82 ± 1.99 | 17.34 ± 1.06 | 30.62 ± 1.96 |
| PASS (Zhu et al., 2021) | - | - | 24.11 ± 0.48 | 39.25 ± 0.90 | 18.73 ± 1.43 | 32.01 ± 1.68 |
| FeTrIL (Petit et al., 2023) | - | - | 30.97 ± 0.90 | 36.51 ± 1.67 | 25.70 ± 0.61 | 39.54 ± 1.19 |
| LDC (Gomez-Villa et al., 2024) | 44.09 ± 0.22* | 54.67 ± 0.41* | 34.20 ± 0.70 | 46.80 ± 1.10 | 24.75 ± 0.10* | 38.71 ± 1.10* |
| **LDC+PDR** | **45.88±0.02** | **55.87±0.15** | **37.55±0.12** | **49.77±1.42** | **28.14±0.30** | **42.77±0.92** |

$f(\cdot; \theta)$), we adopt the hyperparameter settings from LDC (Gomez-Villa et al., 2024), including the choice of $\beta$. The adapter $h(\cdot; \psi)$ is implemented as a simple linear layer. It is trained for 20 epochs using both real samples and pseudo samples generated by the decoder $g(\cdot; \phi)$.

**Baselines.** Our prototype update mechanism is plug-and-play and can be seamlessly integrated with most prototype-based CIL methods. Before the PDR stage, any prototype-based method can be chosen to train the encoder. Here we select LDC (Gomez-Villa et al., 2024) as our baseline given its strong empirical performance, training efficiency, and methodological simplicity. A brief overview of LDC is provided below. The encoder training objective of LDC is the same as LwF:

$$\mathcal{L}_{enc} = \frac{1}{|\mathcal{D}_t|} \sum_{(x,y) \in \mathcal{D}_t} \mathcal{L}_{ce}(W_t^\top f(x; \theta_t), y) + \beta \mathcal{L}_{kl}(W_t^{p\top} f(x; \theta_t), W_{t-1}^\top f(x; \theta_{t-1})),$$

where $\mathcal{L}_{ce}$ denotes the cross-entropy loss, and $\mathcal{L}_{kl}$ denotes the KL divergence. The classifier $W_t$ is divided into $W_t^p$ representing class weights of $\mathcal{C}_{1:t-1}$, and $W_t^c$ representing class weights of $\mathcal{C}_t$. Moreover, LDC use Nearest Class Mean (NCM) classifier (Goswami et al., 2024; Gomez-Villa et al., 2024). Specifically, LDC classifies the test images based on their distance to all the class prototypes as $y^* = \arg\min_{y \in \mathcal{C}_{1:t}} ||f(x; \theta_t) - \mathcal{P}_{1:t}||$. Same as previous exemplar-free methods (Zhu et al., 2021; 2022; Magistri et al., 2024; Gomez-Villa et al., 2024), we compare LDC+PDR with other state-of-the-art (SOTA) exemplar-free methods, and we do not compare with the exemplar-based methods since they store past task exemplars, resulting in unfair comparison. An asterisk (*) following the data indicates that the results are reproduced by us using the open-source code from the original paper, since the results of those metrics are not reported in previous works. Other results are all reported by the original paper. We report the mean and standard deviation of each metric using 5 different seeds, shuffling the classes in order to reduce the bias induced by the choice of class ordering (Kirkpatrick et al., 2017; Magistri et al., 2024), and the referred results of previous methods are also based on this setting.

The dash ('-') indicates that experiments for that setting were not implemented for the method. The best result for each metric is marked in red, and the second-best result is marked in blue.

## 5.2 EXPERIMENTAL RESULTS

**Cold Start.** In Table 1, Table 2 and Table 3, we compare LDC+PDR with SOTA exemplar-free approaches on CIFAR100, TinyImageNet and ImageNet100. LDC+PDR significantly outperforms the previous SOTA methods in both metrics across most settings of the considered datasets.

Table 3: Evaluation on ImageNet100 with the protocol that equally splits classes into $T$ tasks.

| Method | $T=5$ | | $T=10$ | | $T=20$ | |
|---|---|---|---|---|---|---|
| | $A_{last}$ (%) | $A_{inc}$ (%) | $A_{last}$ (%) | $A_{inc}$ (%) | $A_{last}$ (%) | $A_{inc}$ (%) |
| EWC (Kirkpatrick et al., 2017) | - | - | 24.59±4.13 | 39.40±3.05 | 12.78±1.95 | 26.95±1.02 |
| LwF (Li & Hoiem, 2017) | - | - | 37.71±2.53 | 56.41±1.03 | 18.64±1.67 | 40.23±0.43 |
| ABD (Smith et al., 2021) | 51.46 | 67.32 | 35.96 | 57.76 | 22.40 | 44.89 |
| R-DFCIL (Gao et al., 2022) | 53.10 | 68.15 | 42.28 | 59.10 | 30.28 | 47.33 |
| CCIL (Wang et al., 2025) | 56.80 | 70.39 | 46.04 | 62.66 | 34.60 | 49.18 |
| SSRE (Zhu et al., 2022) | - | - | 25.42±1.17 | 43.76±1.07 | 16.25±1.05 | 31.15±1.53 |
| PASS (Zhu et al., 2021) | - | - | 26.40±1.33 | 45.74±0.18 | 14.38±1.22 | 31.65±0.42 |
| FeTrIL (Petit et al., 2023) | - | - | 36.17±1.18 | 52.63±0.56 | 26.63±1.45 | 42.43±2.05 |
| LDC (Gomez-Villa et al., 2024) | 61.36* | 74.77* | 51.40±1.20 | 69.40±0.60 | 32.26* | 51.29* |
| **LDC+PDR** | **64.02±0.02** | **76.44±0.44** | **51.84±1.93** | 68.80±0.80 | **34.78±1.31** | **54.74±4.19** |

Table 4: Sensitivity of performance to the hyperparameter $\alpha$ on CIFAR100 and TinyImageNet.

| Metric | CIFAR100 | | | | TinyImageNet | | | |
|---|---|---|---|---|---|---|---|---|
| | $\alpha = 1$ | 0.2 | 0.1 | 0.05 | $\alpha = 1$ | 0.2 | 0.1 | 0.05 |
| $A_{last}$ | 46.87 | 47.11 | 47.44 | 47.53 | 38.84 | 39.03 | 38.88 | 38.81 |
| $A_{inc}$ | 63.31 | 63.70 | 63.61 | 63.58 | 53.42 | 53.46 | 53.42 | 53.37 |

In Table 1, we present the baseline results and our method on CIFAR100. For $T = 5$, the $A_{last}$ and $A_{inc}$ metrics of our method are 60.53% and 71.80%, which exceeds the best baseline LDC by 1.34% and 0.57% respectively. For $T = 10$, the $A_{last}$ and $A_{inc}$ metrics of our method are 48.14% and 64.00%, which exceeds the best baseline by 2.74% and 3.86% respectively. For $T = 20$, the $A_{last}$ and $A_{inc}$ metrics of our method are 39.80% and 56.70%, which exceeds the best baseline by 3.42% and 2.40% respectively.

In Table 2, our method consistently demonstrates exceptional performance on the TinyImageNet dataset. Our method particularly excels in the setting $T = 10$, where it leads all compared methods by securing the top $A_{last}$ score of 37.55% and the top $A_{inc}$ score of 49.77%. In the setting $T = 5$, our method also achieves the highest $A_{last}$ of 45.88% and $A_{inc}$ of 55.87%. Even in the more challenging scenario $T = 20$, PDR maintains its competitive edge, securing the best $A_{last}$ result of 28.14%. This robust and often leading performance across various incremental task configurations clearly establishes PDR as a highly effective mechanism for improving prototype-based methods.

As detailed in Table 3, our method, PDR, continues its strong performance by outperforming the majority of baselines under different incremental settings.

**Warm Start.** We evaluated our method under this paradigm on the CIFAR100 dataset, with the comparative results detailed in Table 3. In the 5-task configuration (T=5) on CIFAR100, our method demonstrates a commanding performance, achieving an $A_{last}$ of 62.71% and an $A_{inc}$ of 70.27%. This represents

Figure 3: Evaluation in the Warm Start setting on CIFAR100.

| Method | $T=5$ | | $T=10$ | |
|---|---|---|---|---|
| | $A_{last}$ (%) | $A_{inc}$ (%) | $A_{last}$ (%) | $A_{inc}$ (%) |
| EWC (Kirkpatrick et al., 2017) | 30.47 ± 3.38 | 50.08 ± 1.13 | 21.08 ± 1.09 | 41.00 ± 1.11 |
| LwF (Li & Hoiem, 2017) | - | - | 20.73 ± 1.47 | 41.95 ± 1.30 |
| ABD (Smith et al., 2021) | 50.55 ± 1.14 | 62.40 ± 1.17 | 43.65 ± 2.40 | 58.97 ± 1.87 |
| R-DFCIL (Gao et al., 2022) | 54.76 ± 0.76 | 64.78 ± 1.58 | 49.70 ± 0.61 | 61.71 ± 1.17 |
| PASS (Zhu et al., 2021) | 56.73 ± 0.44 | 65.94 ± 0.68 | 53.42 ± 0.48 | 63.42 ± 0.69 |
| LDC (Gomez-Villa et al., 2024) | 51.14 ± 1.52 | 61.30 ± 1.32 | 45.13 ± 0.02 | 56.83 ± 0.62 |
| **LDC+PDR** | **62.71 ± 0.30** | **70.27 ± 0.65** | **55.01±0.28** | **67.35±0.42** |

a significant lead over all other listed baselines. This outperformance is even more substantial when compared to LDC: LDC+PDR leads LDC by 11.05% in $A_{last}$ and 9.65% in $A_{inc}$ for the 5-task setting. This considerable margin over LDC is also maintained in the 10-task setting, where LDC+PDR shows leads of 9.87% in $A_{last}$ and 10.52% in $A_{inc}$.

**Ablation Study.** PDR's only hyperparameter that needs to be adjusted is $\alpha$ in $\mathcal{L}_{dec}$. In Table 4, we evaluated the results for different $\alpha$ and found that the performance is insensitive to $\alpha$.

# 6 DISCUSSION

**Analysis of Memory Cost.** Our method employs a decoder to generate pseudo exemplars and an adapter to refine prototypes during the training phase of each incremental task. Crucially, these

auxiliary components—the decoder and adapter—are discarded once the adaption of prototypes is complete. Therefore, in any prototype-based method, such as LDC, incorporating the PDR mechanism introduces no additional memory cost.

**Time Cost of PDR Mechanism.** We compare the training time of LDC before and after incorporating the PDR mechanism under identical hardware and experimental settings (CIFAR100 Cold Start $T = 10$). Additionally, we compare both training time and accuracy with other state-of-the-art (SOTA) methods. For each method, we adopt the training hyperparameters that correspond to its best-performing results. As shown in Table 3, LDC+PDR has the same training time as LDC. This is because the Training Decoder step in PDR can be executed in parallel with encoder training, and its runtime is shorter than that of the encoder, resulting in no additional time overhead per session. Moreover, LDC also had the Training Adapter and Re-encoding Prototypes steps, and these steps have low computational complexity and require minimal time. Compared to other SOTA methods, LDC+PDR achieves superior evaluation accuracy while requiring significantly less training time than most of the prototype-based approaches (only longer than SSRE). Importantly, PDR only modifies the training phase and introduces no additional computational cost during inference.

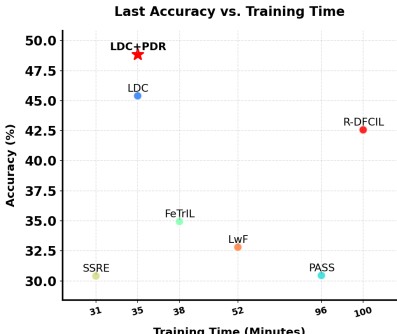

Figure 4: Comparison of the $A_{last}$ (vertical axis) and total training time (horizontal axis, logarithm). All methods are trained on a single RTX 4090 GPU using their open-source implementations.

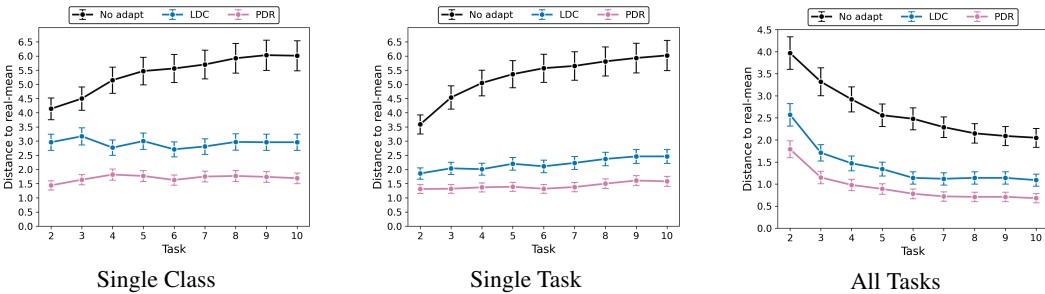

| Single Class | Single Task | All Tasks |

Figure 5: $L_2$ distance between updated prototypes and oracle protoypes after each task on CIFAR100 Cold Start $T = 10$. The leftmost plot (**Single Class**) reports the L2 distance between the stored prototype and the oracle prototype of class 1 from task 1 at the end of each session. The middle plot (**Single Task**) shows the average L2 distance between the stored prototypes and the oracle prototypes of all classes in task 1 at the end of each session. The rightmost plot (**All Tasks**) presents the average L2 distance between all stored prototypes in memory and their corresponding oracle prototypes at the end of each session.

**Quality of Prototype Updating by PDR.** We evaluated the quality of updated prototypes via $L_2$ distance to oracle prototypes (derived from prior class data) on CIFAR100, calculated in the new feature space after each task's encoder training. We compare PDR (adapter with real/pseudo exemplars) against LDC (real exemplars for adaptation) and a No Adapt baseline (unadapted prior task prototypes). The performance hierarchy demonstrates PDR's updated prototypes are significantly closer to ideal oracle prototypes than those from LDC, and vastly superior to unadapted ones, justifying the effectiveness of PDR mechanism.

**Comparison with Deep Generative Replay.** Unlike deep generative replay (DGR) (Shin et al., 2017; Gao & Liu, 2023), our method does not maintain a generative model across sessions, thus incurring far lower memory cost since only class prototypes are stored and the decoder and adapter are discarded once the session ends. Moreover, while DGR trains its generator with both synthetic and real samples, our decoder is trained solely using the old encoder and stored prototypes. Finally, DGR mixes pseudo exemplars generated by the generative model with real exemplars for encoder training, whereas PDR uses the decoder-generated pseudo exemplars solely for prototype updates.

## REPRODUCIBILITY STATEMENT

The code is packaged into a .zip file in Supplementary Material. All major experimental results reported in the paper can be easily reproduced by following the instructions provided in the README file included in the package. The default hyperparameter settings in the released code are consistent with those used for the reported results. In addition, detailed explanations of the experimental setup can be found in Section 5.1, where we provide the values of all hyperparameters used during training. Consulting this section may help in reproducing the results and better understanding our released code.

## ETHICS STATEMENT.

This work adheres to the ICLR Code of Ethics. Our research focuses on algorithmic and methodological contributions in continual learning and does not involve human subjects, sensitive personal data, or information that raises direct privacy or security concerns. The datasets used in our experiments (e.g., CIFAR100, TinyImageNet) are widely adopted public benchmarks released under permissive licenses, and we follow standard usage practices without modification that could introduce ethical risks. The proposed methods are intended for advancing machine learning research and have no foreseeable harmful applications. We are not aware of any conflicts of interest, and the study complies with established principles of fairness, transparency, and research integrity.

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

# Appendix

## THE USE OF LARGE LANGUAGE MODELS (LLMS)

The LLMs are used only to help polishing writing in this work.

