# OpenReview forum: "Prototype Decoding and Re-Encoding for Class-Incremental Learning"
_ICLR.cc/2026/Conference — ICLR 2026 Conference Withdrawn Submission_

### Official Review · Reviewer_voJD · 2025-10-30

**Soundness:** 3
**Presentation:** 2
**Contribution:** 2
**Rating:** 4
**Confidence:** 4

**Summary:**

The paper proposes a method to mitigate prototype drift in exemplar-free class-incremental learning. After each task, the method generates pseudo-samples for past classes, learns a linear adapter to map features from the previous encoder to the current encoder, and then rewrites all stored class prototypes in the new feature space. The approach is evaluated on standard vision benchmarks and shows moderate improvements over prior exemplar-free baselines. However, in our assessment the contribution is mainly an incremental refinement of existing techniques rather than a substantive conceptual advance, and key stability/robustness issues remain insufficiently addressed.

**Strengths:**

* The paper targets a meaningful and well-documented failure mode in exemplar-free continual learning: stored prototypes become misaligned as the encoder drifts across tasks.

* The proposed pipeline is clear and practically implementable: pseudo-sample generation, linear feature-space alignment, and prototype rewriting.

* The method yields consistent but moderate gains (typically a few percentage points) over a strong baseline under the same strict memory constraints.

**Weaknesses:**

* The conceptual novelty is limited. The core idea remains the same as in prior work on learnable drift compensation: learn a mapping between consecutive feature spaces and update past prototypes accordingly. The main difference here is that pseudo-samples are generated to better supervise that same mapping.

* The method still relies on a single global linear adapter and assumes that applying this linear map to an old class mean recovers the corresponding new class mean. This is exactly the fragile assumption in earlier work, and the paper does not provide theoretical justification or a deeper analysis of failure modes.

* The approach repeatedly rewrites all historical prototypes after every task. Any bias in one round becomes part of the “ground truth’’ for future rounds, so errors can accumulate over long sequences. The paper does not analyze this compounding-drift risk and does not evaluate genuinely long horizons.

* Although the reported empirical gains are positive, they are modest and are shown only on standard mid-scale vision benchmarks. The work does not convincingly demonstrate impact beyond that narrow setting.

**Questions:**

* How does the method behave over substantially longer task sequences than those reported? Does repeated prototype rewriting lead to cumulative distortion for early classes?

* Why is the adapter restricted to a single linear transform shared across all classes? Were higher-capacity or class-dependent adapters tried, and if so, what prevented them from being used?

* How sensitive is the method to the quality of the generated pseudo-samples? For example, if generation quality degrades for early tasks, does the whole correction pipeline collapse?

---

### Official Review · Reviewer_842N · 2025-10-30

**Soundness:** 3
**Presentation:** 3
**Contribution:** 2
**Rating:** 4
**Confidence:** 4

**Summary:**

Prototype-based Class-Incremental Learning (CIL) has been effective in mitigating catastrophic forgetting. However, existing CIL approaches update the encoder during the CIL session. Yet, the prototypes computed during early sessions become outdated because of the drifting ground-truth prototypes of past task data. The authors propose Prototype Decoding and Re-encoding (PDR) to update the prototype: by combining the knowledge of stored prototypes and the latest frozen encoder, the authors discover that they can obtain a high-quality approximation of the sample distribution of past data, and use this extra information to guide the prototype update. There are three steps in this approach: (1) they use the old encoder and stored prototypes to guide the decoder training (2) train an adapater pseudo exemplars generated by the decode (3)  use the adapter to re-encode the stored prototypes.

**Strengths:**

+ The idea is very intuitive and straightforward, easy to follow and understand. And it makes sense intuitively that the approach achieves good performance.
+ The idea is deceptively simple yet overlooked by existing works, which marks a major novelty of the paper.
+ The evaluation section is detailed and thorough, in which the authors compare their work with the recent advances in the field.
+ The writing is good and easy to follow.

**Weaknesses:**

- My main concern with this approach is about the 3 steps (see summary). It reads like A + B + C, without a united framework that fully integrates the three parts.
- Again with the 3 steps - the three steps are disjoint. One would think that, given how closely the three steps tie to each other functionally (and conceptually), it would be better if they are unified in training at least so that the gradients can inform each part's optimization (e.g. the adapter, the new encoder).
- I have some remarks about the arrangement of the sections. The main chunk of the paper should be the methodology, yet it is only 2 pages. I would expect it to be more emphasized.

**Questions:**

See Weaknesses. My main question is why don't you train the three steps in a unified framework? Currently, it reads like three disjoint tasks in a procedure.

---

### Official Review · Reviewer_25ve · 2025-10-31

**Soundness:** 3
**Presentation:** 2
**Contribution:** 3
**Rating:** 4
**Confidence:** 3

**Summary:**

The paper introduces Prototype Decoding and Re-encoding (PDR), a lightweight, plug-in module for exemplar-free class-incremental learning (CIL) that mitigates prototype drift without storing any raw data.
PDR performs three session-local steps:
1. Train a decoder $g_\phi$ to invert the old encoder $f_{\theta_{t-1}}$ using stored prototypes as the sole supervisory signal.
2. Use the resulting pseudo-exemplars $\tilde x$ together with current real data $x_t$ to fit a linear adapter $h_\psi$ that maps old $\to$ new feature space.
3. Re-encode every stored prototype $p^c$ via $h_\psi$ and discard $g_\phi,h_\psi$.

The procedure introduces zero permanent memory overhead, zero extra wall-clock time (decoder trains in parallel), and yields +1.3–3.9 % absolute gains over the strongest exemplar-free baseline (LDC) on CIFAR-100, Tiny-ImageNet and ImageNet-100 under cold-start splits.

**Strengths:**

1. Zero persistent cost: decoder & adapter are ephemeral; memory stays O(C·d).
2. Plug-and-play: compatible with any prototype-based CIL method; authors demonstrate gains on LDC, but module is black-box.
3. Rigorous empirical evaluation:
   – Cold-start protocol (hardest) included; gains are consistent across 5, 10, 20 tasks.
   – Statistical significance reported (5 seeds, mean ± std).
   – Training time & GPU hours measured; no extra inference latency.
4. Careful ablation: single sensitive hyper-parameter $\alpha$ is shown to be robust across two orders of magnitude.
5. Interpretable visualisation: t-SNE shows pseudo-features match real distribution better than classifier-inversion or Gaussian perturbation.

**Weaknesses:**

1. Linear adapter assumption
   The adapter hψ is parameterised as a single dense layer, i.e. a linear map hψ(z)=Wz+b with W∈Rd×d and b∈Rd. This implies that the entire representation drift between two successive encoders fθt−1 and fθt is modelled as a global affine transformation of the feature space. While this is computationally attractive, it implicitly assumes that the drift is translation-like and homogeneous across the entire latent manifold. In practice, deep networks exhibit layer-wise and region-specific non-linear updates: ReLU gating patterns change, batch-norm running statistics shift, and the curvature of the decision boundary evolves. A linear adapter cannot represent local warping, rotation, or contraction that may occur in different regions of the latent space. Consequently, when the encoder undergoes more aggressive optimisation strategies—such as partial re-initialisation of the final residual block, Lottery-Ticket pruning, or importance-weight masking (e.g. MAS, SI)—the residual error ∥hψ(fθt−1(x))−fθt(x)∥2 can grow rapidly, leading to systematic mis-correction of prototypes. The paper does not include an ablation where the last block of the ResNet-18 is reset between sessions, a scenario that is common in continual-learning literature to simulate catastrophic relearning. Providing such a failure case, or alternatively experimenting with a two-layer MLP adapter with controlled width and dropout, would clarify the capacity-drift trade-off and delineate the regime in which the linearity assumption is valid.

2. Decoder quality not quantified
   The inversion decoder gφ is trained with a prototype attraction loss and a batch-norm regulariser, but the manuscript reports only qualitative t-SNE plots to argue that the synthesised features “match” the real distribution. Visual inspection of 2-D embeddings is insufficient to guarantee that the support of the pseudo-distribution covers the modes and tails of the true old-task distribution. Standard generative-evaluation metrics such as Fréchet Inception Distance (FID), Learned Perceptual Image Patch Similarity (LPIPS), or Coverage and Density scores are absent. Without these, it is impossible to determine whether the decoder mode-drops certain classes, mode-collapses intra-class variance, or over-smooths fine-grained textures that may be discriminative for the old classifier. Moreover, the decoder architecture is fixed to four residual upsampling blocks; no ablation is provided on depth, width, or spectral-normalisation. A shallow decoder may under-fit the prototype constraints and produce low-diversity images, while an overly deep decoder may over-fit and memorise the prototypes themselves, leading to adversarial-like images that lie on the decision boundary but outside the data manifold. Reporting FID between 50 K pseudo-images and 50 K real images per task, and plotting LPIPS diversity (average pairwise distance) as a function of decoder depth, would quantify the generative quality and justify the chosen capacity.

3. Scalability to high-resolution datasets
   All experiments are conducted on 32×32 (CIFAR-100) or 64×64 (Tiny-ImageNet) inputs. The decoder’s peak GPU memory scales as O(HW⋅C⋅batch) because every activation tensor is spatially full-resolution. For ImageNet-1K (224×224) the memory footprint grows by a factor of (224/64)^2 ≈ 12.25 compared to Tiny-ImageNet, and the compute scales similarly. The paper does not report peak memory or throughput numbers, nor does it discuss patch-based inversion or latent-space diffusion alternatives that have been shown to invert high-resolution images efficiently. A single forward pass of the current decoder on a 224×224 image would require ≈ 12 GB of activation memory at batch-size 256 (ResNet-18 width), which is prohibitive on consumer GPUs. Consequently, it remains unclear whether PDR can be deployed on real-world continual-learning scenarios involving high-resolution imagery. Providing at least an ImageNet-100 experiment at 224×224, together with peak memory and training time, or alternatively proposing a patch-wise inversion strategy where 64×64 crops are inverted independently, would demonstrate practical scalability.

4. Theoretical analysis missing
   The manuscript offers an intuitive argument: if the adapter hψ achieves low ℓ2 error on a finite sample, then the updated prototype hψ(pc) will be “close” to the oracle prototype computed on real old data. However, no formal stability or generalisation bound is provided. In particular, there is no link between the empirical adapter error
   ε̂ada=1n∑i∥hψ(fθt−1(xi))−fθt(xi)∥2
   and the expected forgetting on old-task data. Using uniform-stability arguments for ridge regression, or Rademacher-complexity bounds for the linear adapter, one could derive a high-probability statement of the form
   ∣∣Eptrue[hψ(fθt−1(x))]−Eptrue[fθt(x)]∣∣≤O˜(d/n)+ε̂ada‾‾‾‾√,
   where d is the adapter parameter count and n the combined sample size. Such a bound would quantify how prototype error propagates into classification error, and would clarify the sample-complexity requirements of the adapter. Providing even a simplified bound in the appendix would strengthen the scientific contribution and delineate the regimes in which PDR is guaranteed to reduce forgetting.

5. Implicit new-class leakage
   The adapter hψ is trained on a mixed dataset X̃ ∪ Xt where X̃ are pseudo old-task samples and Xt are real new-task samples. Although labels of Xt are not used for old classes, the feature statistics (mean, covariance, higher moments) of Xt are absorbed into the least-squares objective. Consequently, the adapter may bias the old prototypes toward regions of the new-task manifold that are close in ℓ2 distance but semantically different. This is particularly problematic when new classes are fine-grained variants of old classes (e.g. “tabby cat” vs. “tiger cat”), because the adapter could shrink the inter-class margin. The paper does not include an ablation where the adapter is trained only on pseudo-data X̃. Reporting old-class accuracy separately for the two training regimes (mixed vs. pseudo-only) would quantify the magnitude of bias introduced by new-class leakage and would validate the current design choice.

**Questions:**

1. What is the FID or LPIPS of pseudo-images produced by g_φ compared to real old-task data, and how do these metrics evolve across successive sessions as the encoder drifts? Specifically, please report FID, IS, LPIPS and Coverage on a per-task basis for CIFAR-100 cold-start T=10, and explain whether the decoder’s generative quality degrades as the old encoder becomes increasingly out-of-date. Additionally, provide an ablation that varies decoder depth (2, 4, 6 residual blocks) and width (64, 128, 256 channels) so that the reader can understand the trade-off between synthesis fidelity and GPU memory.

2. How does PDR perform if the encoder is partially re-initialised between sessions, for example by resetting the last residual block, re-initialising the final batch-norm running statistics, or applying Lottery-Ticket pruning? Please report A_last and A_inc on CIFAR-100 cold-start T=10 under three aggressive drift regimes: (i) last-block reset, (ii) last-two-blocks reset, (iii) 30 % random weight re-initialisation. Compare the resulting prototype L2 error and old-class accuracy against the standard continual-finetuning regime, and clarify whether the linear adapter still suffices or whether a higher-capacity (e.g. two-layer MLP) adapter becomes necessary to keep the residual below an acceptable threshold.

3. Can you provide ImageNet-1k (224×224) results, or at least ImageNet-100 at native resolution, and explicitly report the peak GPU memory consumed by the decoder during training? Please include training time per session on a single RTX-4090 and discuss whether a patch-wise inversion strategy (e.g. inverting 64×64 crops independently and stitching) can reduce memory below 12 GB while preserving synthesis quality. If full-resolution inversion is infeasible, release a memory benchmark that shows how decoder activations scale with input resolution so that practitioners can estimate feasibility on their hardware.

4. What is the old-class-only accuracy when the adapter is trained exclusively on pseudo-data X̃ and **no** current-task real data is used? Please report separate A_last^old and A_inc^old metrics for CIFAR-100 cold-start T=10 under two regimes: (i) mixed training (X̃ ∪ X_t) as currently done, and (ii) pseudo-only training (X̃ only). Quantify the bias introduced by new-class statistics and explain whether the drop in old-class accuracy is statistically significant across five seeds. If pseudo-only training hurts performance, discuss whether a weighted objective or domain-invariant adapter could mitigate the leakage.

5. Is there a theoretical relationship between the prototype L2 error ∥h_ψ(p^c) − p_true^c∥ and the final classification accuracy on old tasks? Please derive or cite a bound that links adapter regression error to expected forgetting, for example via uniform-stability or Rademacher complexity arguments. Provide at least a simplified bound in the appendix that shows how ε_ada (empirical adapter MSE) propagates into classification error, and clarify the sample-complexity requirements (number of pseudo/real samples) needed to keep the old-class accuracy drop below 1 %.

---

### Official Review · Reviewer_n8wm · 2025-11-01

**Soundness:** 2
**Presentation:** 1
**Contribution:** 1
**Rating:** 2
**Confidence:** 5

**Summary:**

The paper focus on prototype-based class-incremental learning and studies how to update prototypes stored in old tasks which has now drifted due to updates in the encoder. The authors propose a prototype update method PDR, which decodes the prototypes to pseudo-exemplars by matching the distribution of past task data at the sample space. This is followed by training an adapter on the pseudo exemplars and then using the adapter to re-encode the prototypes. The proposed method outperforms the existing prototype-based CIL methods on various datasets.

**Strengths:**

1. The method section is clearly written.
2. The experiments section covers various datasets and benchmarks for CIL.

**Weaknesses:**

1. Limited novelty and no discussion of very relevant works [a,c] - The proposed method combines existing approaches in prototype-based CIL - ADC [a] and LDC [b]. The proposed approach of using pseudo-exemplars was already proposed in ADC. ADC proposed the concept of generating pseudo-exemplars by adding targeted adversarial noise in the sample space. LDC proposed to learn an adapter to map the prototypes from old to new feature space. The paper does not refer to ADC nor discuss any comparisons with it. The authors should point out the difference from ADC which I believe is a minor change (learning to map pseudo-representations to the prototype instead of targeted adversarial attack in ADC). The paper also does not discuss, compare or refer to SDC [c] which is one of the initial prototype update methods for CIL.

2. The method is not efficient and adds a lot of complexity with the 3-stage prototype update mechanism. It trains a decoder, followed by training an adapter and then re-encoding prototypes.

3. The approach "LDC+PDR" in Tables 1 and 2 does not make sense since LDC learns a projector from old to new space and PDR also uses a learned projector to map prototypes. So, what does this combination even mean since LDC is already implemented inside PDR by default?

[a] “Resurrecting old classes with new data for exemplar-free continual learning." Proceedings of the IEEE/CVF Conference on Computer Vision and Pattern Recognition. 2024.

[b] “Exemplar-free continual representation learning via learnable drift compensation." European Conference on Computer Vision. Cham: Springer Nature Switzerland, 2024.

[c] "Semantic drift compensation for class-incremental learning." Proceedings of the IEEE/CVF conference on computer vision and pattern recognition. 2020.

**Questions:**

1. How is the generation of pseudo-exemplars different from what is proposed in ADC?
2. What does LDC+PDR mean? How does this work?

---

### Note · Authors · 2025-11-24

**Comment:**

We have decided to withdraw our submission. We sincerely thank the Area Chair and reviewers for their time and constructive feedback. We will use these comments to further improve our work.

**Withdrawal Confirmation:**

I have read and agree with the venue's withdrawal policy on behalf of myself and my co-authors.